# EmoTour: Estimating Emotion and Satisfaction of Users Based on Behavioral Cues and Audiovisual Data

**DOI:** 10.3390/s18113978

**Published:** 2018-11-15

**Authors:** Yuki Matsuda, Dmitrii Fedotov, Yuta Takahashi, Yutaka Arakawa, Keiichi Yasumoto, Wolfgang Minker

**Affiliations:** 1Graduate School of Information Science, Nara Institute of Science and Technology, Nara 630-0192, Japan; takahashi.yuta.to2@is.naist.jp (Y.T.); ara@is.naist.jp (Y.A.); yasumoto@is.naist.jp (K.Y.); 2Fellow of Japan Society for the Promotion of Science, Tokyo 102-0083, Japan; 3RIKEN, Center for Advanced Intelligence Project AIP, Tokyo 103-0027, Japan; 4Institute of Communications Engineering, Ulm University, 89081 Ulm, Germany; dmitrii.fedotov@uni-ulm.de (D.F.); wolfgang.minker@uni-ulm.de (W.M.); 5ITMO University, Saint Petersburg 197101, Russia; 6JST Presto, Tokyo 102-0076, Japan

**Keywords:** ubiquitous computing, emotion recognition, satisfaction estimation, wearable computing, dialogue systems, smart tourism, smart cities

## Abstract

With the spread of smart devices, people may obtain a variety of information on their surrounding environment thanks to sensing technologies. To design more context-aware systems, psychological user context (e.g., emotional status) is a substantial factor for providing useful information in an appropriate timing. As a typical use case that has a high demand for context awareness but is not tackled widely yet, we focus on the tourism domain. In this study, we aim to estimate the emotional status and satisfaction level of tourists during sightseeing by using unconscious and natural tourist actions. As tourist actions, behavioral cues (eye and head/body movement) and audiovisual data (facial/vocal expressions) were collected during sightseeing using an eye-gaze tracker, physical-activity sensors, and a smartphone. Then, we derived high-level features, e.g., head tilt and footsteps, from behavioral cues. We also used existing databases of emotionally rich interactions to train emotion-recognition models and apply them in a cross-corpus fashion to generate emotional-state prediction for the audiovisual data. Finally, the features from several modalities are fused to estimate the emotion of tourists during sightseeing. To evaluate our system, we conducted experiments with 22 tourists in two different touristic areas located in Germany and Japan. As a result, we confirmed the feasibility of estimating both the emotional status and satisfaction level of tourists. In addition, we found that effective features used for emotion and satisfaction estimation are different among tourists with different cultural backgrounds.

## 1. Introduction

Due to the ubiquity of smart devices, including smartphones and wearables, people can find various and helpful pieces of real-time living environment information, such as about the weather and roadway traffic. Moreover, some of the recently emerged systems account for user context, e.g., recording what/where users are doing at that moment. To design more context-aware systems, psychological user context (e.g., emotional status) needs to be taken into account since it differs across users, and even for the same user at different times. In this study, we focus on the tourist domain. It is a typical use case with high demand for context awareness. In fact, the emotional status and satisfaction level of tourists are susceptible during sightseeing; hence, observing emotional feedbacks is useful for providing context-aware tourist guidance.

The aim of our study is to estimate the emotional status and satisfaction level of users susceptible to rating their activities (e.g., tourists during sightseeing). Recently, various approaches have been proposed to understand users’ emotional status and satisfaction level in order to use them for consumer services. The most widely used approaches to collect satisfaction level of users are online user reviews and questionnaires [1,2]. User reviews are also used in consumer services, such as the rating systems of TripAdvisor [3], Yelp [4], and Amazon [5]. However, keeping users motivated to regularly write reviews is difficult, especially with medium rating values, which leads to the risk of biased distribution of review ratings. To provide reliable information for other users, it is necessary to collect quantitative data without user reviews. Though many studies have tried to estimate the emotional status of users with methods based on audiovisual-data analysis [6,7,8,9,10,11,12], the accuracy of emotion recognition in outdoor places, such as tourist sights, tends to be worse due to the inclusion of environmental noise in audiovisual data [13,14,15]. In recent studies, the range of modalities has been expanding to physiological features (e.g., body movement, eye gaze) [16,17,18,19,20,21,22,23,24], and a fusion of them might help with even outdoor estimation [17,25,26].

In this article, we propose a tourist emotion- and satisfaction-estimation system named EmoTour. EmoTour employs several kinds of modalities taken from actions that tourists naturally and unconsciously do during sightseeing. Since tourists often naturally record videos or take pictures by themselves (e.g., a selfie), and also unconsciously behave (e.g., walk around, gaze at an object) during sightseeing, audiovisual data and behavioral cues are used as modalities. In our previous paper, we have already confirmed that behavioral cues have a relationship with the emotion and satisfaction of tourists [27]. With EmoTour, we have built a model for estimating tourists’ emotion and satisfaction during sightseeing by fusing features derived from each modality.

The main contributions of this paper are the following:First, we propose a new model for quantitatively estimating both the emotion and satisfaction of tourists by employing multiple modalities obtained from unconscious and natural user actions. To avoid the potential risk of biased ratings in a user review for satisfaction-level estimation, and enable emotional-state estimation at an actual sightseeing situation, we employ the combination of behavioral cues and audiovisual data collected by an eye-gaze tracker, physical-activity sensors, and a smartphone. In detail, the following high-level features were derived from each modality and fused to build a final classifier: eye movement, head tilt, and footsteps from behavioral cues; and vocal and facial expressions from audiovisual data. We argue that our scheme can build the model without dependence on any extra tasks for users.Second, we evaluated our model through experiments with 22 users in a tourist domain (i.e., in a real-world scenario). As the experimental fields, we selected two touristic areas, located in Germany and Japan, which have completely different conditions. We evaluated the emotion estimation model through a three-class classification task (positive, neutral, negative) using unweighted average recall (UAR) score as a metric, and achieved up to 0.48 of UAR score. Then, we evaluated the satisfaction estimation model through a 7-level regression task (0: fully unsatisfied–6: fully satisfied) using mean absolute error (MAE) as a metric, and achieved up to 1.11 of MAE. In addition, we found that effective features used for emotion and satisfaction estimation are different among tourists with different cultural background.

The rest of the paper is organised as follows. Section 2 introduces the current status of related studies and services, and defines the challenges that should be overcome in our study. Section 3 considers the approach for estimating user emotion and satisfaction, and explains the concrete workflow of our approach and the modalities used. Section 4 describes the methodology of user emotion and satisfaction estimation including feature extraction and modality fusion. Moreover, we evaluate our method through real-world experiments in Section 5, and discuss the contribution and limitation of our method in Section 6. Finally, conclusions and suggetions for future work are given in Section 7.

## 2. Related Work and Challenges

Due to the high demands of context-aware systems, especially in the tourist domain, there are many studies focused on environmental sensing to collect real-time information of tourist sights [28,29]. On the other hand, the estimation of the psychological user context, which may be used for providing appropriate information based on the situation, has not yet been deeply tackled in spite of its importance.

Our motivation was the enhancement of a context-aware tourist-guidance system by introducing psychological context estimation in addition to existing environmental sensing technologies. In the following sections, we describe related works that widely include other domains, then clarify the objective and challenges of our study, and introduce our preliminary work.

### 2.1. Estimation of Emotional Status

Emotion recognition has been a hot topic for many research areas and for several years now due to the high demand for context-aware systems, such as spoken-dialogue systems. However, there are not many studies targeting the tourist domain yet.

In emotion recognition, audio- and/or visual-based approaches are popular fields in the field of dialogue systems and human–computer interaction [30]. In laboratory (indoor) conditions, existing audio-based emotion-recognition systems that use a deep neural network have achieved great performance [6,7]. Quck et al. proposed an audio-based emotion-recognition system [8]. They built a dialogue system on mobile devices, and achieved around 60% recall score for four affective dimensions. Tarnowski et al. proposed an approach based on facial movements [9]. They obtained good classification accuracy of 73% for seven facial expressions. Moreover, they mentioned that head movements (orientation) could significantly affect extracting facial-expression features. Aiming at higher accuracies, bimodal emotion-recognition methods, combining audio and visual features, were also proposed [10,11], and they achieved better accuracy (e.g., 91% for six emotion classes). However, in outdoor conditions, the accuracy of emotion recognition tends to be worse due to the inclusion of environmental noise in audiovisual data. According to Emotion Recognition in the Wild Challenge 2017 (EmotiW), the accuracy of emotion recognition is up to 60% for seven emotion classes [13,14,15]. Since tourist sights may have noisier environments, we should take such environmental conditions into account to estimate the emotional status of tourists.

To infer the emotion of a person, the unconscious behavior of humans may be a clue as well. Shapsough et al. described that emotions could be recognised by using typing behaviour on the smartphone [16]. This approach used a machine-learning technique and induced high accuracy on emotion recognition, yet it is not feasible to frequently ask users to type on their smartphone during a sightseeing tour. Resch et al. proposed an emotion-collecting system for urban planning called Urban Emotions [17]. The paper describes that wrist-type wearable devices and social media were used for emotion measurements. Since this approach relies on an assumption that posts on social media are written in situ, it has the problem of spatial coverage for collecting data. In recent large social media services (e.g., Twitter, Facebook), users cannot attach exact location data to a post with default settings, which makes UrbanEmotions difficult to collect comprehensive data. However, it also suggested that body movement can be used for recognising emotion.

Moreover, recent studies in the field of emotion recognition focus on expanding the range of modalities and combining them. Ringeval et al. proposed to introduce physiological features in addition to audiovisual ones, and to build a multimodal system that relies on their combination [12]. As physiological features, an electrocardiogram and electrodermal activity were used. Physiological features provided lower performance and weaker correlation than audiovisual ones with continuous emotional labels, but helped to increase the overall performance of a multimodal system. Many studies also introduce physiological features: heart-related (electrocardiogram, heartbeat), skin- and blood-related (electrodermal activity, blood-pressure), brain-related (electroencephalography), eye-related (eye gaze, pupil size), and movement-related (gestures, gyroscopic data) [17,18,19,20,21,22,23,24], and, in many cases, improvement of accuracy was observed, even in outdoor conditions [17,25,26].

### 2.2. Estimation of Satisfaction Level

In current consumer services, such as TripAdvisor [3], Yelp [4], and Amazon [5], online user reviews and questionnaires are still widely used to collect the satisfaction level of users. TripAdvisor [3] in particular uses a five-star rating system and comments from tourists as user reviews about sightseeing spots. To guarantee quantity and quality of voluntary reviews, it is essential to provide a motivation to contributors. However, keeping users motivated to regularly write reviews is difficult, especially with medium rating values, because many people do not like to post a review without any external incentives when they felt “there’s nothing special”. It means there is a risk of skewing the evaluation due to an imbalance of reviewers’ distribution.

Many studies also adopt questionnaire-based surveys for measuring tourist satisfaction [1,2]. Fundamentally, several hundred samples (respondents) are required to produce reliable data. However, because the questionnaire-based method relies on the manual tasks of a human, it has problems in sustainability and the spatial coverage of the survey. It also has the same risk as the method based on user reviews and ratings.

### 2.3. Objective and Challenges

Our objective was to determine the quantitative emotional status and satisfaction levels of users to design more intelligent and reliable guidance systems. However, through the investigation of current studies and services, we found several problems (e.g., biased reviews, spatial coverage of evaluation, accuracy of estimation) for applying existing techniques to real conditions of use.

From this background, the main challenge of our study was to establish a state-of-the-art method for estimating user emotion and satisfaction by fusing audiovisual data and various sensor data specialising in user behavior. In the following sections, we describe the design and implementation of a method for estimating the emotional status and satisfaction level of tourists, and provide deeper evaluation and discussion through real-world experiments.

### 2.4. Preliminary Work

As our preliminary work, we found a correlation between user psychological context (emotional status, satisfaction level) and user unconscious behavior [27]. Then, we proposed a basic setup for estimating the emotional user status, and confirmed the feasibility of the concept of our study through a small experiment in real-world conditions [31]. In this paper, we explain the method for estimating not only the emotional status but also the satisfaction level in detail, and evaluate both methods through experiments with additional participants (nine people added, 1.7 times the scale). Moreover, we analyzed the effects of differences in cultural backgrounds of tourists on the accuracy of estimation, and discuss future perspectives.

## 3. Proposed Approach and Workflow

Our approach was designed for the tourist domain, where users are walking on a path through different sightseeing areas. To estimate the emotion and satisfaction of tourists during sightseeing, we focus on various actions that tourists unconsciously and naturally do during sightseeing. For example, tourists might approach a scenic place or work of art, stop, and gaze at it, potentially take selfie photos, or send video messages to their friends. The accumulation of these natural actions should be linked to the emotional status and satisfaction level that they feel there. Through a preliminary study, we have already confirmed that several actions (eye gaze and head/body movement) have a relationship with the emotion and satisfaction of tourists [27]. Hence, we propose an approach to observe such natural actions of tourists and estimate the tourist emotion/satisfaction while performing them.

Figure 1 shows the workflow of our whole system for collecting the data of tourist actions and labels. The overview of each step is as follows:
**Step 1**—**Split the whole tour into sessions**Before starting sightseeing, we split the whole tour into small periods (sessions) that included at least one sight each. We assumed that a tourist typically requests guidance information for each sightseeing spot.**Step 2**—**Sensing and labeling**Tourists could freely visit sights while equipped with wearable devices that continuously recorded their behavior during the whole sightseeing. At the end of each session, they gave small amounts of feedback about the latest session by recording a selfie video. We assumed that recording a video serves as a means of interacting with dialogue systems or sending a video message to their friends. They also manually input their current emotional status and satisfaction level as a label. Then, they repeated the same procedure for each of the tour sessions.**Step 3**—**Building the estimating model**The tourist emotion- and satisfaction-estimation model was built based on tourist behavior, audiovisual data, and labels.

In the following sections, we describe the details of the modalities and labels.

### 3.1. Modalities

To perform an emotion estimation in the tourist domain, we used multimodal features: audiovisual data (vocal/facial expressions) and behavioral cues (eye and head/body movement data). Since tourists often take videos or photos, e.g., a selfie, audiovisual data could be used for our study. However, accuracy may have been low due to environmental issues in outdoor places, as mentioned in Section 2. Hence, we additionally used the features extracted from various tourist behaviors that happen unconsciously during sightseeing. The sense of sight is one of the most important sensory systems in sightseeing, and it can be tracked as eye movement using existing wearable devices and technologies. Moreover, due to the directivity on sensory systems (e.g., hearing, sight), head and body movements may be affected by them. Thus, we used head and body movements as features in addition to eye movement.

In our study, we used the three devices shown in Figure 2 to record features in real time: an Android smartphone (GPS-data, audiovisual data), a mobile eye-tracking headset Pupil with two 120 Hz eye cameras [32] (eye gaze, pupil features), and a sensor board SenStick [33] mounted on an ear of the eye-tracking device (accelerometer, gyroscope).

### 3.2. Labels

To represent the psychological context of tourists, we employed two types of metrics: emotional status and satisfaction level. We collected these data as labels by using the Android application shown in Figure 3. Tourists could manually enter the ratings of the session at the end of each session. The details of each metric are described as follows:**Emotional** **status**To represent the emotional status of tourists, we adopted the two-dimensional map defined on Russell’s circumplex space model [34]. Figure 4 shows the representation of the emotional status. We divided this map into nine emotion categories and classified them into three emotion groups as follows:
**Positive**  :  Excited (0), Happy/Pleased (1), Calm/Relaxed (2)  **Neutral**   :  Neutral (3)  **Negative** :  Sleepy/Tired (4), Bored/Depressed (5), Disappointed (6),        Distressed/Frustrated (7), Afraid/Alarmed (8)
As a side note, Russell’s model [34] is mainly employed for the time-continuous annotation of audiovisual databases that we used to build pre-trained models in Section 4.**Satisfaction** **level**To represent the satisfaction level of tourists, we used the Seven-Point Likert scale which the Japanese government (Ministry of Land, Infrastructure, Transport, and Tourism) uses as the official method. Tourists could choose their current satisfaction level between 0 (fully unsatisfied) and 6 (fully satisfied). A neutral satisfaction level is 3 and it should approximately represent the state of the participant at the beginning of the experiment.

## 4. Methodology of Tourist Emotion and Satisfaction Estimation

In this section, we describe the methodology of estimating emotion and satisfaction of tourists. Figure 5 depicts the scheme of our method, which consists of three stages. The first stage is Data Collection described in Section 3. The second stage is Feature Extraction, described in Section 4.1, where we preprocessed the collected raw data and extracted several features for each modality. The final stage is Modality Fusion described in Section 4.2, where several modalities were combined to build the final classifier of tourist emotion and satisfaction estimation.

### 4.1. Preprocessing and Feature Extraction

The raw data from each modality cannot be directly used to build tourist emotion and satisfaction estimation model. Hence, the methods of data preprocessing and feature extraction explained in Section 4.1.1 (behavioural cues) and Section 4.1.2 (audio-visual data) are applied to each modality.

#### 4.1.1. Behavioral Cues—Eye-, Head-, and Body-Movement Features

Eye-movement features were extracted using the Pupil Labs eye tracker [32]. We used *theta* and *phi* values, which represent a normal pupil as a 3D circle in spherical co-ordinates (Figure 6a). Hence, we could only use two variables to describe the position of pupils and, thus, the eye gaze. Note that the raw values of eye-movement data differ across users and depend on the physical setting of camera and eye peculiarity. The eye-gaze data were analyzed using the following methodologies:
**F1:** **Intensity of eye movement**Minimum and maximum values for *theta* and *phi* were calculated for each participant; eight thresholds (10%–90%, 10% step, except 50%) were set for the range [min, max] as shown in Figure 6b, and then used to count the percentage of time outside each threshold per session. In total, 16 features were used.  **F2:** **Statistical features of eye movement**Average and standard deviation of *theta* and *phi* were calculated for a small window of recorded data and the values corresponding to the same session were averaged. The following window sizes were used: 1, 5, 10, 20, 60, 120, 180, and 240 s with the offset of $\frac{1}{3}$ of the window size. In total, 64 features were used.

Then, head and body movement features were extracted using the inertial-sensor (accelerometer and gyroscope) values of the SenStick [33]. Both sensors have three axes: the X-axis, Y-axis, and Z-axis. Head- and body-movement data were analyzed using the following methodologies:
**F3:** **Head movement (head tilt)**As a head movement, head tilt was derived using gyroscope values. The average μ and the standard deviation σ of the gyroscope values were calculated for each participant. Then, the upper/lower thresholds ψ were set with the following equations (Equations (Equation 1) and (2)). The parameter *a* represents the axis of the gyroscope.
(1)$$\begin{array}{c} \psi_{upper,a}=\mu_{a}+2\sigma_{a}, \end{array}$$
(2)$$\begin{array}{c} \psi_{lower,a}=\mu_{a}-2\sigma_{a}. \end{array}$$
Finally, head tilt (looking up/down, right/left) was detected using threshold ψ. In our condition, the Y-axis indicates a looking-up/down motion, and the Z-axis indicates a looking-left/right motion. Since the duration of each session was different, we converted these data to several features: head tilt per second; and average and standard deviation of the time interval looking at each direction. In total, 23 features were used.  **F4:** **Body movement (footsteps)**Footsteps are analyzed with a method based on the approach of Ying et al. [36]. First, the noises of accelerometer values were removed by applying a Butterworth filter with 5 Hz cutoff frequency. Then, high-frequency components were emphasised through the differential processing shown in Equation (Equation 3). The parameter x(n) represents the accelerometer value at index *n*.
(3)$$\begin{array}{c} y\left(n\right)=\frac{1}{8}\left\{2x(n)+x(n-1)-x(n-3)-2x(n-4)\right\}. \end{array}$$
Furthermore, the following integration process (Equation (Equation 4)) smoothed the accelerometer values, and small peaks of them were removed. In our condition, *N* was chosen to be 5 empirically. Since the sensor position was different from the original method in our condition, we used a modified parameter.
(4)$$\begin{array}{c} y\left(n\right)=\frac{1}{N}\left\{x(n-(N-1))+x(n-(N-2))+\cdots +x(n)\right\}. \end{array}$$
Finally, footsteps were extracted by counting local maximum points. As features, we used footsteps per second, and average and standard deviation of a time interval for each step. In total, five features were used.

#### 4.1.2. Audiovisual Data—Vocal and Facial Expressions

Audio features (vocal expressions) were extracted with openSMILE software [37]. They consisted of 65 low-level descriptors (LLDs) of four different groups (prosodic, spectral, cepstral, and voice quality) and their first-order derivatives (130 features in total) used in ComParE challenges since 2013 [38]. Window size was set to 60 ms, and window step size was set to 10 ms, resulting in feature extraction on overlapping windows with a rate of 100 Hz.

As video features (facial expressions), action units (AUs) were extracted with OpenFace [39,40], an open-source toolkit. AUs describe specific movements of facial muscles in accordance with the facial action coding system (FACS) [41,42], e.g., AU-1 means an action of “raising up the inner brow.” We used the following 17 AUs, which can be extracted using OpenFace: 1, 2, 4, 5, 6, 7, 9, 10, 12, 14, 15, 17, 20, 23, 25, 26, 28, 45. The procedure of AU extraction is shown in Figure 7. First, the movie file is converted to a sequence of images, and facial landmarks are detected for each frame. Then, the face images are clipped out from the original frames because it expects that background objects and random people are partially captured in movie data in outdoor places. Finally, AUs are recognized by fusing several features taken from a clipped image for each frame of an original movie. In total, 18 AU-related features were extracted.

Due to the lack of data for training a decent audiovisual-based emotion-recognition system, in this study we used models that are trained in advance by utilizing several existing corpora of emotionally rich interactions: the RECOLA (Remote COLlaborative and Affective interactions) database [12], SEMAINE (Sustained Emotionally coloured MAchine-human Interaction using Nonverbal Expression) database [43], CreativeIT database [44], and RAMAS (The Russian Acted Multimodal Affective Set) database [45]. As corpora have different annotation rates, they were brought to the same data frequency to be able to share the same prediction models. The least frequency of 25 Hz, presented in RECOLA, was used for the remaining corpora.

The models are based on recurrent neural networks with long short-term memory (RNN-LSTM) and built in a way to consider the particular amount of context (7.6 s) in accordance to our previous study [46], as it shows better results. Networks were comprised of two hidden LSTM layers of 80 and 60 neurons with ReLU (Rectified Linear Unit) activation function, respectively, each followed by a dropout layer with a probability of 0.3. The last layer has one neuron with linear activation function for regression tasks (databases: RECOLA, SEMAINE, CreativeIt) and six neurons with softmax activation function for the classification task (database RAMAS). We used RMSProp [47] as an optimizer with a learning rate of 0.01. For regression tasks, we utilized a loss function based on the concordance correlation coefficient that takes into account not only the correlation between two sets, but also the divergence, being not immune to biases. For the classification task, we used cross-categorical entropy.

After feeding the features extracted from the audiovisual data of our experiment to the trained model, for each time step we obtained the prediction in arousal and valence if we used the regression models, and probabilities of particular emotion if we used the classification models.

Models that are trained on the additional corpora cannot be directly used for emotional-status estimation in the context of our method due to the following reasons:Labels differ from those collected through our system in range and dimensions, i.e., they are on the arousal–valence scale instead of emotions for regression tasks, and an emotion set for a classification task does not match with ours. They are time-continuous, i.e., each value represents the emotional state for one frame of the audiovisual data, though we had one label per each session.

Taking these differences into account, predictions should be generalised and adapted to our method without losing valuable information. To achieve it, we took simple functionals (*min*, *max*, *mean*) from dimensional labels, i.e., arousal and valence as well as mean prediction scores for categorical emotions separately for each session, and merged them into feature vectors. Thus, we had high-level predictions from earlier trained models as features in our system.

For each modality, we used a simple feed-forward neural network with one hidden layer to make unimodal prediction from high-level features.

### 4.2. Modality Fusion

To build our final tourist emotion- and satisfaction-estimation system, we combined predictions based on our features, described in Section 4.1, on two levels: feature and decision. During the experiment, some problems with the devices and the data-collection process occurred that led to some data missing. The final classifier should be robust and able to work with incomplete feature sets. On the decision-level fusion, it was achieved by applying linear models, where the final label is assigned, based on a linear combination of existing lower-level predictions. On the feature level, such feature sets were filled with zeros.

## 5. Experiments and Evaluation

### 5.1. Overview of Real-World Experiments

We conducted experiments in real-world conditions to evaluate the tourist emotion- and satisfaction-estimation method. As the experimental fields, we selected two tourist areas depicted in Figure 8 that have completely different conditions. The first one is the centre of Ulm, Germany. The sights in this area include particular buildings as well as walking routes with high tourist value (e.g., the Fisherman’s Quarter). The sights are surrounded by common city buildings and may be crowded depending on the time. The approximate length of the route is 1.5 km, divided into eight sessions. The second area is Nara Park, the historic outskirts of Nara, Japan. The route through the area includes many scenic and religious buildings (temples and shrines) that are located in nature, and has no distraction from the sights included in the sessions. The approximate length of the route is 2 km, divided into seven sessions.

Participants were asked to follow prepared routes and take as much time as they needed to see the sights. During the sessions, we recorded the data in real time according to Figure 1. At the end of each session, participants were asked to provide small feedback and labels for emotion and satisfaction level.

In total, we conducted our experiment with 22 participants, and collected 183 sessions’ data. The distribution of participants was the following: age range—22–31 years old (average age is 24.3); nationalities—12 Japanese, 10 Russian; gender—17 males, 5 females. In addition, 17 and 5 people went sightseeing to the tourist area located in Germany and Japan, respectively. Most of them were real tourists (e.g., short-term international students or new students, visitors), and hence they were not familiar with the experimental fields.

To collect the labels, the emotional status for each session during sightseeing could be measured only by the participants themselves. Due to the natural impression that sightseeing is something interesting, labels collected from tourists tend to be imbalanced. In our experiments, the distribution of labels was as shown in Figure 9a, and the ratio of each emotion group was: positive: 71.0%, neutral: 17.5%, negative: 11.5%. To manage this condition, we used UAR as the performance metric. The same imbalance is presented for the satisfaction level as shown in Figure 9b. Then, the relationship between emotion categories (groups) and satisfaction level is shown in Figure 10a,b, and they suggest a moderate positive correlation (r=0.644), which proves the agreement between labels.

### 5.2. Results

To evaluate our emotion- and satisfaction-estimation system, we conducted a series of reproducible experiments. To provide a fair comparison of different modalities or fusion methods, we fixed random seeds, resulting in consistent sample shuffling. The results for uni- and multimodal emotion and satisfaction estimation are presented in Table 1. We applied two different fusion approaches: feature-level fusion, where we built one model on merged feature vectors from corresponding modalities; and decision-level fusion, where we used a linear combination of prediction scores from unimodal models to set the final label.

We evaluated the performance of our emotion estimation system through a classification task of three emotion groups: Positive (0–2), Neutral (3), and Negative (4–8). Due to imbalanced label distribution, we used UAR as the performance metric. This ranged from 0 to 1 (the higher the better) and, for three-class classification problems, it had a chance level of 0.33. The results (Table 1) show that our system performed at up to 0.48 of a UAR score in three-class (all emotions) emotion-estimation tasks. At the unimodal level, we found a performance of 0.45 using head/body-movement features (F3, F4). This suggests that there are implicit connections between emotional status and features, calculated for these modalities. Then, the highest performance of 0.48 was shown with all features (decision-level fusion), and it proves that combining behavioral cues and audiovisual data is useful to estimate the tourists’ emotions.

Then, to evaluate the performance of satisfaction estimation, we derived the MAE between estimated value and label. The MAE score can be any positive value, and 0 represents a perfect match. Table 1 shows that our system performed at a low MAE up to 1.12 in seven-level satisfaction-estimation tasks. The highest performance was shown with all features (decision-level fusion) as with emotion-estimation tasks.

In our experiment, participants could be roughly divided into two almost equal groups by their nationality: Japanese (12 people) and Russian (10 people). It was proven by previous studies that different culture groups express emotions differently in ways and intensity [48,49,50]. To see whether the difference in cultural background affected the accuracy of estimation in our study as well, we conducted the same modeling procedures, described above, with two culturally differentiated groups of participants. Then, to evaluate the performance of emotion and satisfaction estimation, the UAR score and MAE were derived, respectively.

The results are shown in Table 2, and they suggest that the impact of cultural difference exists, especially on emotion-estimation tasks. For Japanese tourists, the result shows that audiovisual data showed a better UAR score of 0.45 than behavioral cues (0.42) at the unimodal level. Then, we confirmed that their decision-level fusion showed the highest UAR score of 0.47. Interestingly, we found that there was an opposite characteristic for Russian tourists. The highest UAR score of 0.57 was shown by behavioral cues; in contrast, the low score of 0.34–0.41 was shown by audiovisual data, and video-based models performed almost at a chance level.

These results suggest that we need to take the effects of nationalities or cultural differences into account in order to generalize the model of our system. As future work, we will expand the nationalities of tourists, and build a general model using them in consideration of their cultural background. In addition, we aim to investigate the effects of other attributes of tourists, such as gender and age.

## 6. Discussion and Limitations

### 6.1. Feasibility of Our Proposed System

The results from Section 5.2 prove that tourists’ emotion and satisfaction estimation can be extracted by our proposed system to a certain degree. In our study, modality fusion showed a better result compared to unimodal systems for estimating both emotional status and satisfaction level. Especially combining behavioral features at a feature-level often improved the performance of emotion estimation, compared to eye- and head-based feature alone. The possible reason for this is that, according to the process of data collection and human movements, eye-gaze and head movement are connected to each other: a human moves them both while exploring an environment, usually replacing a significant eye movement with a slight head movement. The combination of these modalities at a feature level allows the system to simultaneously utilize information from both sources, which is not possible to do at a decision level.

### 6.2. Imbalance of Labels

Studies related to emotion estimation often suffer from the subjectivity of labels. In our study, we had an exceptional case of subjectivity, as emotion and satisfaction can be measured only by the participants themselves and not by any third parties, such as an annotator. An additional limitation was brought by the domain of our research—tourism. As the main idea was to measure people’s first impression, they could not participate twice in the same experiment, and should not be familiar with the experimental field. This means that we could not ask local citizens to participate in an experiment, which constrained the range of potential candidates to a very narrow group. These conditions resulted in a data shortage, complicating the model training stage, affecting the general performance and statistical stability of results.

Natural perception of sights as something interesting and the general conditions of an experiment led to a great imbalance of emotional labels. Some emotion groups are predefined to be almost empty a priori because these emotions can be caused by sightseeing, e.g., distressed or afraid. Even after dividing emotions into three groups (positive, negative, and neutral) unequally, we had 71% of positive samples. Many of these problems and limitations can be partially overcome by increasing the amount of data, which would make the system more stable and robust.

### 6.3. Limitation of Data Sources

To realize the proposed system, we need to collect several data sources, such as eye-gaze data, head motions, and selfie videos. However, there is a limitation in collecting such data in real-life conditions. Although eye-tracking devices are becoming smaller and cheaper year by year, it will take more time for them to be commonly and frequently used. However, head-motions can be measured because JINS MEME [51] with electro-oculography (EOG) and a six-axis inertial measurement unit (IMU) are already being sold on the market, and many people are using them in their daily life. In case of using selfie videos, we must take it into account that the emotional status on such videos may be exaggerated. One of the most possible shifts may be done from natural emotions to acted ones. If so, it is possible as future work to utilize existing databases for emotion recognition to improve the performance of the system.

The aim of our study, as our first attempt, was to reveal what kind of data are needed for estimating the emotional status and satisfaction level of tourists. Hence, we employed all kinds of conceivable modalities in this paper, but we do not assert that all the modalities are required. Through the experiments, we have confirmed that various modalities and their fusion can be used for estimating emotional status and satisfaction level. We also found that there is no notable difference in the estimation performance at the unimodal level; however, performance can improve by combining them in several ways. This suggests that our proposed method does not rely on a specific modality or combination. In the current situation, not many tourists take a selfie video that can be used as one of the data sources in our system. However, even without videos, our proposed method can perform at a certain level. Of course, if the selfie video becomes common, like selfie photos, performance can be improved.

### 6.4. Future Perspectives

The result of this paper provides a baseline performance for estimating the emotional status and satisfaction level of tourists. Several ways can be considered to improve performance. One is to widely explore other available modalities and their combinations. For example, foot motion and direction might be used for estimating the degree of tourist interest during sightseeing. Another is to analyze the transition of emotional status and satisfaction level of tourists during a session. Since emotional status and satisfaction level can drastically change even in the same session, this transition process might also be a valuable clue to estimate the tourists’ status at the end of the session.

## 7. Conclusions

To design a more context-aware system, psychological user context should be constantly taken into account. In this study, as a typical use case, we selected the tourist domain, and aimed to estimate the emotional status and satisfaction level of tourists during a sightseeing tour based on their unconscious and natural actions, e.g., selfie videos, body movements, etc. We proposed a tourist emotion- and satisfaction-estimation method by fusing several modalities. To build the model, four kinds of modalities were employed: behavioral cues (eye and head/body movement) and audiovisual data (vocal/facial expressions). Through experiments in the real world with 22 tourists, we achieved up to 0.48 of an unweighted average recall score in the three-class emotion estimation task, and up to 1.11 of mean absolute error in the seven-level satisfaction estimation task. In addition, we found that effective features used for emotion and satisfaction estimation are different among tourists with a different cultural background. As future work, we will expand the nationalities of tourists and build a general model including consideration of cultural background.

## Figures and Tables

**Figure 1 sensors-18-03978-f001:**
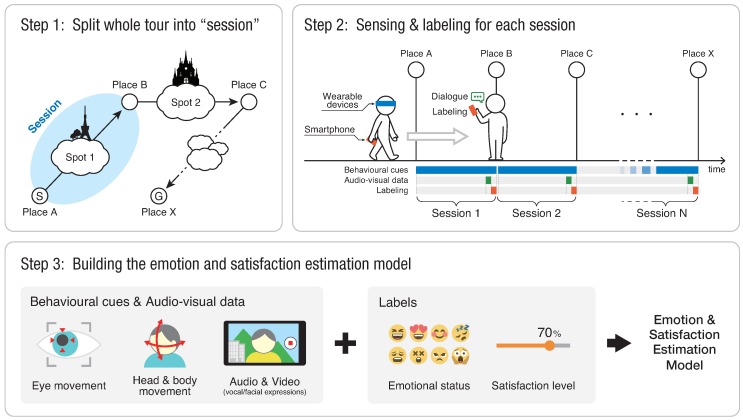
Workflow to estimate tourist emotion and satisfaction level.

**Figure 2 sensors-18-03978-f002:**
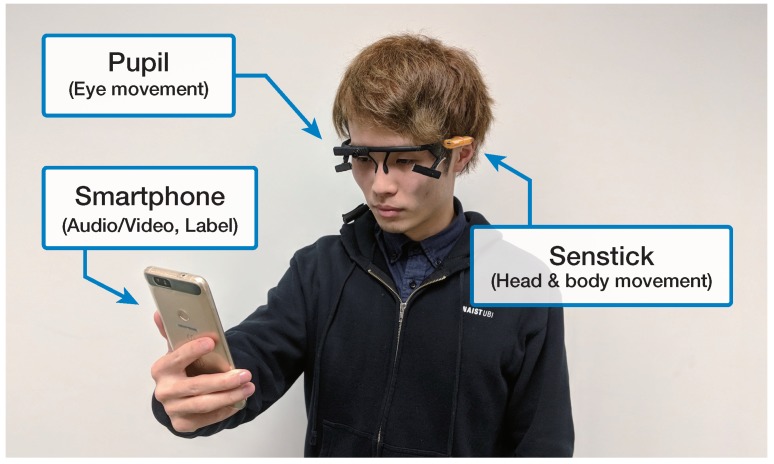
Devices for data collection during sightseeing: pupil [32], SenStick [33], and smartphone.

**Figure 3 sensors-18-03978-f003:**
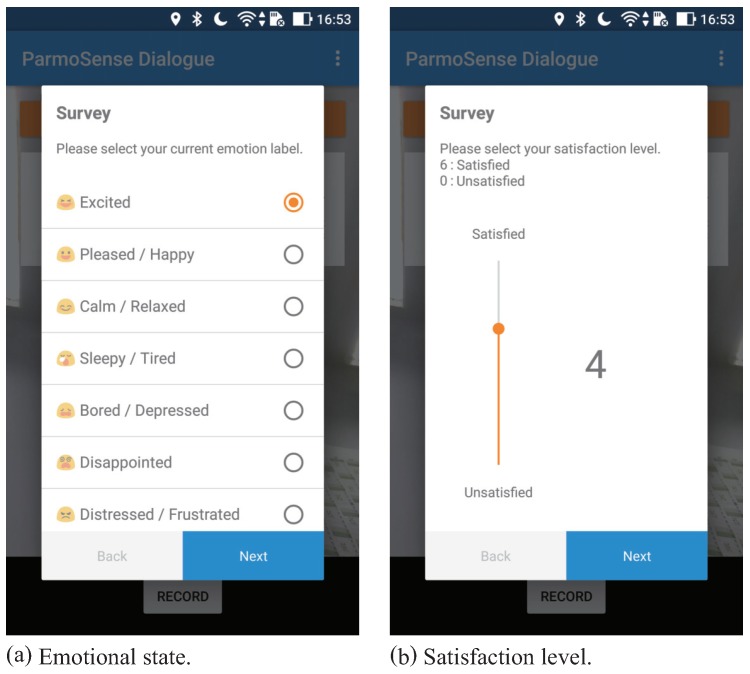
Smartphone application for collecting labels from tourists.

**Figure 4 sensors-18-03978-f004:**
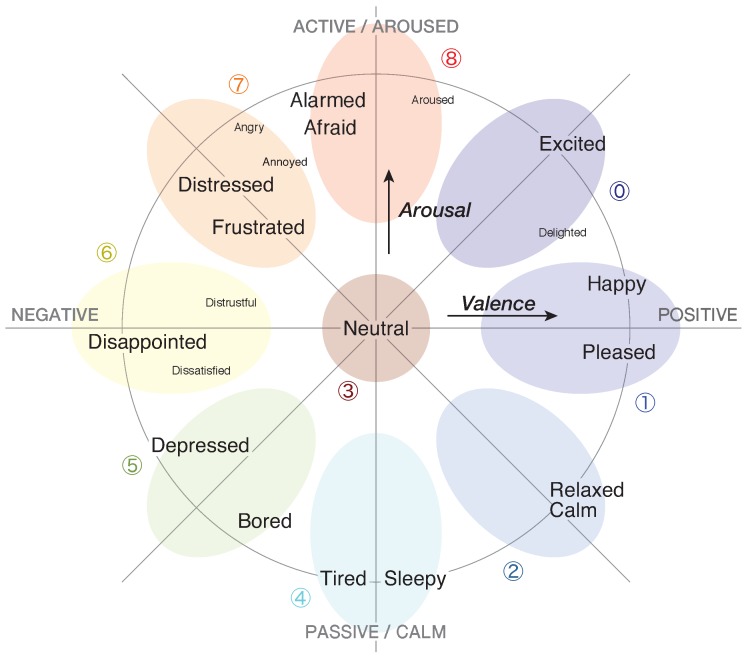
A two-dimensional emotion status model. Figure is taken from References [34,35].

**Figure 5 sensors-18-03978-f005:**
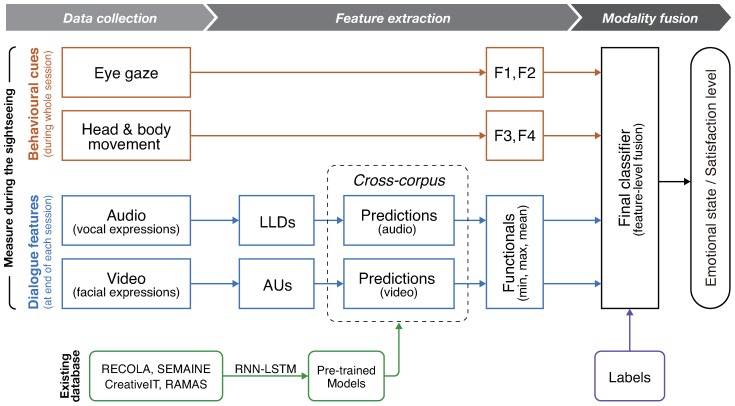
Scheme of tourist emotion and satisfaction estimation with modality fusion. LLDs: low-level descriptors of prosodic, spectral, cepstral, and voice quality, AUs: action units for describing facial expressions, F1: intensity of eye movement, F2: statistical features of eye movement, F3: head movement (head tilt), F4: body movement (footsteps), RNN-LSTM: recurrent neural network with long short-term memory.

**Figure 6 sensors-18-03978-f006:**
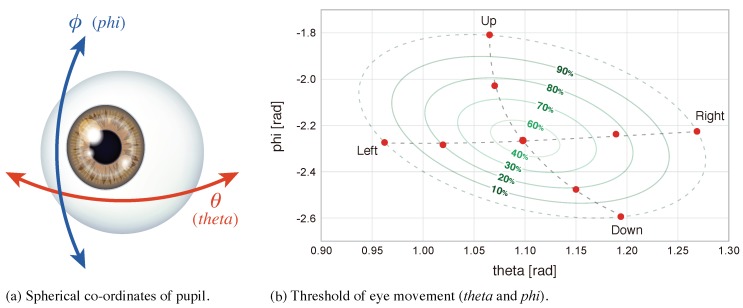
Representation of eye movement.

**Figure 7 sensors-18-03978-f007:**
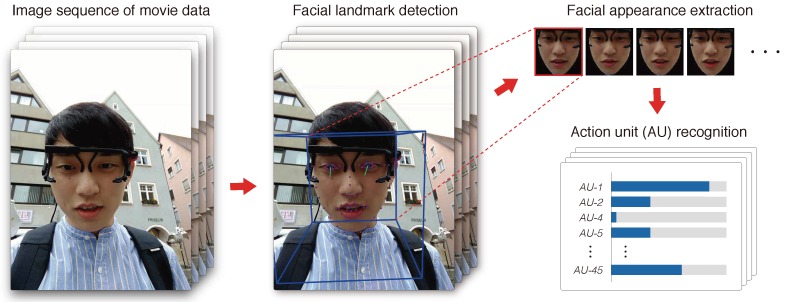
Visual features (action units) extraction using Openface [39,40].

**Figure 8 sensors-18-03978-f008:**
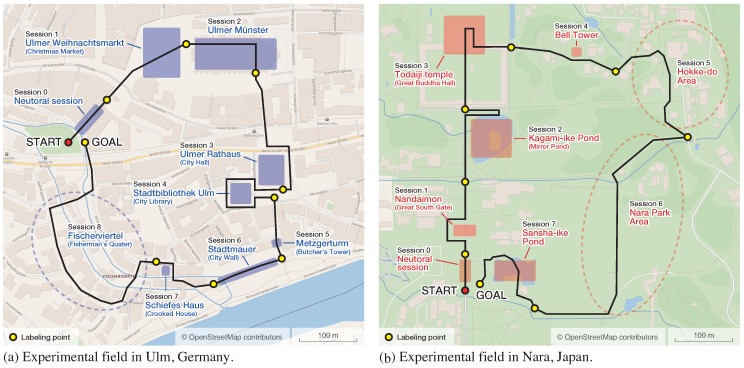
Experimental fields.

**Figure 9 sensors-18-03978-f009:**
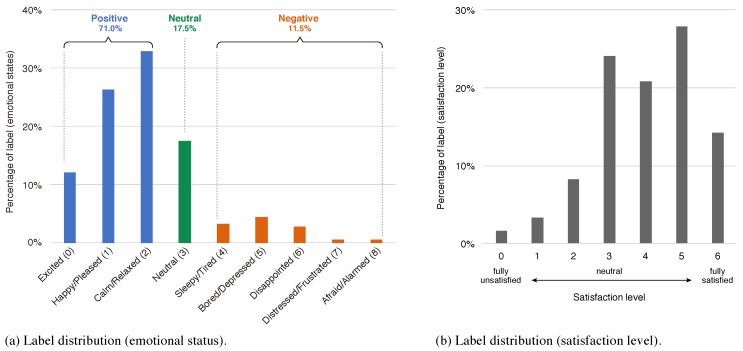
Label distribution.

**Figure 10 sensors-18-03978-f010:**
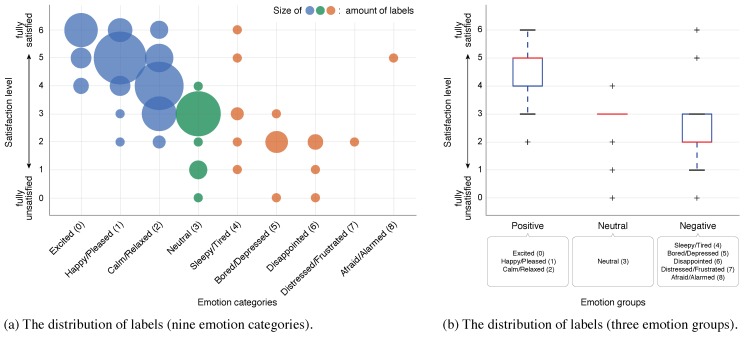
Relationship between emotional status and satisfaction level.

**Table 1 sensors-18-03978-t001:** Performance of uni- and multimodal tourist emotion and satisfaction estimation. The best performances are highlighted with bold text.

Modality	Emotion (Unweighted Average Recall: UAR)	Satisfaction (Mean Absolute Error: MAE)
Eye movement (F1, F2)	0.401	1.238
Head/body movement (F3, F4)	0.434	1.230
**Behavioral cues (eye + head/body movement)**	0.458	1.265
Audio (vocal expressions)	0.386	1.208
Video (facial expressions)	0.411	1.198
**Audiovisual data (audio + video)**	0.414	1.194
**Feature-level fusion**	0.428	1.311
**Decision-level fusion**	**0.484**	**1.110**

**Table 2 sensors-18-03978-t002:** Performance of tourist emotion and satisfaction estimation (by nationality of participants). The best performances are highlighted with bold text.

Modality	Emotion (UAR)		Satisfaction (MAE)	
	Japanese	Russian	Japanese	Russian
Eye movement (F1, F2)	0.438	0.426	1.045	1.345
Head/body movement (F3, F4)	0.417	0.438	1.314	1.290
**Behavioral cues (eye + head/body movement)**	0.415	**0.576**	1.099	1.347
Audio (vocal expressions)	0.447	0.372	1.093	1.304
Video (facial expressions)	0.463	0.346	1.100	1.300
**Audiovisual data (audio + video)**	0.445	0.417	1.067	1.300
**Feature-level fusion**	0.423	0.507	1.190	1.420
**Decision-level fusion**	**0.473**	0.496	**1.000**	**1.157**

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
