# Peer review of "EmoTour: Estimating Emotion and Satisfaction of Users Based on Behavioral Cues and Audiovisual Data"

_sensors, 2018, doi:10.3390/s18113978_

Round 1

Reviewer 1 Report

The authors present the design of a context-aware system for detecting emotional state of a tourist during sightseeing.

They claim that the novelty of including the psychological user context is a substantial factor for providing useful information in appropriate timing. A

Through a study carried out with a tourist guidance prototype system, they aim to estimate the emotional status and satisfaction level of tourists using behavioral cues, such as

eye and head/body movement, and audio-visual data.

In the next please find some details of the review:

-The abstract includes some information about the results that should not be there. For instance some results are presented saying that the recall is 0.48 or up to 1.11 of mean absolute error but the experiments are not known by the reader and thereof are difficult to understand at this point of the article. They have to be omitted and also the abstract should be a bit shorter and more concrete about the objectives and main context rather than in presenting what is done that should be in the introduction. Also there are some improvement in the presentation such as change line 3“(e.g, emotion)” by “and this emotional state”, remember that the abstract has to be concrete and define well the main goal.

-The introduction does not include a broad state of the art about the topic of the paper. The state of the art is briefly explained in 50 lines and all the references are included within a range (line 38 [2-12]) but without explained them and without justifying why there are not valid. There is not any other reference to similar works or explanation what are the advantages of the proposal compared to them. There should be more literature and better explained.

-In the introduction, there is a part for main contributions (line 49-68) and is hard to see the difference between contribution 1 and contribution 2. It seems that they are the same but differently explained. They should be joined or clarify their differences. Moreover in the last of the three it is included the value of the errors obtained before it has been explained how did you get it. It should be remove the exact value from here by a phrase indicating the improvement at quality level.

-In line 78 section 6 is missing.

-In section 2.1  line 95 there are some literature that claim up to 92%, a deeper study of the literature should be done (the authors also should check if part of this sections with related with the state of the art should go in the introductions for better understanding and clarification of the context of the work).

-In section 2.1 paragraph 112-119 as indicated in the previous point there are some works that get better results (up to 94% for valence and arousal recognition; and 100% for happiness, 72.33% for surprise, 96.67% for anger, 79.22% for fear, 96.11% for

disgust, and 66.67% for sadness)  Please check P. C. Petrantonakis and L. J. Hadjileontiadis, “Emotion Recognition from Brain Signals Using Hybrid Adaptive Filtering and Higher Order Crossings Analysis,” IEEE Transactions on Affective Computing, vol. 1014 1, no. 2, pp. 81–97, 2010.  Or  Y.-P. Lin, C.-H. Wang, T.-L. Wu, S.-K. Jeng, and J.-H. Chen, “EEG based emotion recognition in music listening: A comparison of schemes for multiclass support vector machine,” in ICASSP, 2009.

-line 121 the reference to the study that proofs that sentence should be included.

-line 124 it may also be included other similar dataset such as amazon 5-stars reviews http://jmcauley.ucsd.edu/data/amazon/

-In line 147 the authors claim that they already have done the estimation of satisfaction and here as well, if this is an incremental work to a previous one it should be clarify by indicating the differences and objectives of this paper otherwise it should be explained better the relation with [13].

-in line 200 why the authors did not use other emotion classification like Ekman's? or Plutchik's? It should be explained, at least one reason why that one was chose.

-in line 206 Why the authors didn’t use the standard 5-starts rating?

-In figure 5 the RNN that are introduced later and used for video recognition is not included. It should be included in this diagram due to it contains the global vision of the system.

-Paragraph 261-265 how the audio features have been extracted is not fully explained, where the 130 features come from? Are the mean of the different windows (feature step sizes)?

-Paragraph 282-286 a better explanation of the RNN would be needed to be able to reproduce the experiments and for clarification of the paper.

-Paragraph 290-296 is hard to understand it, please rewrite it.

-The results obtained are not as good as presented though there are good enough to think that the prototype could be improve and be usable in a real industrial application. That should be highlighted in the discussion including the way for achieving it.

Reviewer 2 Report

This paper reported a design and experimentation on using several means to capture the emotional state towards objective determination of how a tourist thinks about a place that he/she visits. The methods include eye gaze, head and body movement, and audio and video data. The authors compared the classification result for single modality and multimodality scenarios. 

The sensor signal processing and sensor fusion part are meritable and could be used in scenarios beyond what the authors intended. 

However, for the purpose as stated in the paper, I do not think the proposed method is practical, and it may encounter serious usability issues:

1. It is not conceivable for a tourist to wear some special device to track gaze and head movement.

2. While taking selfie pictures is common, it is far less common to take selfie videos. Furthermore, the nature of the videos, even if they are taken as expected, the emotions shown in the videos may not reflect their true emotion at all. Social convention would make one smile and appear to be happy in the video. How can anyone use such videos to reliably detect the true emotional state?

As such, I encourage the authors to reposition the work and focus on the sensor signal processing and sensor fusion for more liable emotion detection, and consider other use cases for their method. I see some research contributions in the paper, but I do not think using the method to objectively determine the emotional state for tourists is not feasible. 

Round 2

Reviewer 1 Report

I reviewed the manuscript and the authors accomplished with all my reviews. I recommend to accept the paper in its current state.

Reviewer 2 Report

I appreciate that the authors have sincerely attempted to address all the issues I raised. Even though I do not agree all their opinions, I think the presented work could be valuable for the readers. Therefore, I recommend to accept the paper.